# Surgical Treatment of a Retropharyngeal Abscess in a Japanese Black Cow

**DOI:** 10.3390/vetsci9080446

**Published:** 2022-08-20

**Authors:** Shoichi Okada, Kim Sueun, Ryosuke Ichikado, Kohei Kuroda, Yoshiyuki Inoue, Yoshiki Nakama, Hiroyuki Satoh, Reiichiro Sato

**Affiliations:** 1Graduate School of Medicine and Veterinary Medicine, University of Miyazaki, 1-1 Gakuen Kibanadai-nishi, Miyazaki 889-2192, Japan; 2Faculty of Agriculture, University of Miyazaki, 1-1 Gakuen Kibanadai-nishi, Miyazaki 889-2192, Japan; 3Nishimoro Western Branch, Miyazaki Agricultural Mutual Aid Association, 1321-1 Hosono, Kobayashi, Miyazaki 886-0004, Japan

**Keywords:** cattle, drooling, dysphagia, endoscopy, trocar, ultrasound, X-ray, wheezing

## Abstract

**Simple Summary:**

Causes of inappetence and dysphagia in cow include swallowing foreign bodies, such as a magnet or a sharp object, or trauma to the posterior pharyngeal area during oral administration. The cow in this case had formed a large abscess in her pharynx that obstructed the entrance to her trachea, although the cause was unknown. By using ultrasound equipment to check the thin walls of the abscess and the distribution of blood vessels, a hole was drilled in the appropriate area, which allowed the internal pus to drain and heal. Ultrasound diagnostic equipment is useful in diagnosing abscesses deep inside the body.

**Abstract:**

A 17-month-old Japanese Black cow presented with inappetence, wheezing, dysphagia, and drooling. Radiography and ultrasonography revealed an oval, dorsal, pharyngeal mass, with an internal horizontal line demarcating the radiolucent area from the radio-opaque area. Upper airway endoscopy revealed pus-like deposits in the dorsal nasal passage, hyperemia, and edema in the dorsal pharynx, leading to swelling and airway obstruction. Manual palpation, after sedation, revealed a thickened mass surface, which was difficult to rupture with manual pressure. After inserting a linear sonographic probe through the mouth to establish the vascularity surrounding the mass and to identify a relatively thin-walled area, a trocar was pierced into the mass under endoscopic guidance, and the opening was enlarged manually. The mass was filled with stale pus-like material, which was removed manually. The abscess cavity was washed using saline and povidone–iodine. Day 1 post-surgery, dysphagia and wheezing disappeared. Day 16 post-surgery, endoscopy showed significant improvement in the airway patency. One year postoperatively, the owner reported that the cow had an uneventful recovery. For deep abscesses, such as bovine pharyngeal abscesses, it is important to perform a preoperative transoral Doppler ultrasonography to assess the vascularity and thickness of the abscess wall for safe trocar insertion and abscess drainage.

## 1. Introduction

Wheezing is typically caused by stenosis of the upper airways. In cattle, it may be caused by sinusitis or obstruction of the nasal cavity (for example, a foreign body or congenital cyst), pharynx (most commonly an abscess), larynx (for example, necrotizing laryngitis), or trachea (for example, bovine herpesvirus 1) [1].

Abscesses within the pharyngeal wall, causing airway obstruction, develop owing to upper respiratory diseases that occur sporadically in cattle [2]. They may also be caused by the insertion of a foreign body, such as a magnet or a sharp object, or trauma to the posterior pharyngeal area during oral administration [3].

Previous studies recommend surgical treatment of pharyngeal abscesses [2,3,4]. However, the present case differed from those previously reported, as the space between the nasopharyngeal and mesopharyngeal regions was narrow, allowing only the passage of a single human hand. Insertion of a scalpel or other sharp instruments could have damaged the surrounding tissues. Therefore, a trocar was inserted into the mass in the appropriate position, guided by preoperative intraoral ultrasonography and endoscopy to assess the mass vascularity, peripharyngeal area, and thickness of the abscess wall. A positive postoperative outcome was achieved. This article presents the first report of a bovine pharyngeal abscess treated with a trocar.

## 2. Case Presentation

A 17-month-old Japanese Black cow presented with fever and inappetence. A referring veterinarian treated her with oxytetracycline, marbofloxacin, and aqueous dexamethasone; however, the symptoms persisted. The cow’s owner was referred to the Miyazaki University Veterinary Teaching Hospital for further diagnosis and treatment.

On admission, the cow was well-developed, weighing 378 kg, although she experienced some weight loss. Physical examination revealed respiratory abnormalities. Her body temperature was 39.5 °C, heart rate was 108 beats per minute, respiratory rate was 28 breaths per minute, and her neck was stretched forward with labored breathing. Marked wheezing was heard in the upper airways, which was more pronounced during inspiration. The nostrils were mildly moist, and profuse drooling was observed. The patient showed some interest in hay and pellet feed; however, she experienced dysphagia and vomiting. 

## 3. Investigations

### 3.1. Hematology and Blood Chemistry Analyses

Blood samples were obtained from the jugular vein using disposable, anticoagulant-free syringes, EDTA tubes, and a plain vacutainer (Venoject^®^, Terumo Corp., Hyogo, Japan). Complete blood counts were analyzed using an automated cell counter (PocH-100iV DiffSysmex Corp., Tokyo, Japan), and blood chemistry was analyzed using Dri-Chem NX700 (FUJIFILM Medical Co., Ltd., Tokyo, Japan), within 30 min of sample collection. 

The results of the complete blood count were in the normal range (white blood cell count: 8000 cells/μL, red blood cell count: 10.9 × 10^4^ cells/μL, and thrombocytes: 70.0 × 10^4^ cells/μL). The serum total protein concentration was slightly elevated (8.2 mg/dL; reference range: 6.5–7.5 mg/dL).

### 3.2. Radiographic Examination

In the orthostatic position, without sedation, horizontal radiography of the head was performed with a mobile X-ray unit (Sirius Star Mobile, FUJIFILM Healthcare, Tokyo, Japan) centered on the pharynx; it showed a mass with increased soft tissue density in the dorsolateral part of the pharynx. The mass was oval, measuring approximately 12 × 11 cm, with an internal line demarcating the lower radio-opaque region from the upper radiolucent region, indicating the presence of fluids (Figure 1).

### 3.3. Ultrasonography

Ultrasonography (7.0–8.0 MHz variable linear probe, iViz air, FUJIFILM Healthcare, Tokyo, Japan) was performed without sedation, from the left side of the cow, using the trachea and epiglottic cartilage as landmarks. The interior of the mass was hypoechogenic, though not fluid (Figure 2).

### 3.4. Endoscopy

Endoscopy (9.2 mm video scope, VES Be-one, Olympus Medical Systems Corp., Tokyo, Japan) of the upper airways was performed without sedation. It showed pus-like deposits on the dorsal nasal septum, hyperemia, and edema of the dorsal pharynx. The swelling was pronounced, obstructing the airways and preventing further progression of the endoscope to examine the larynx (Figure 3).

The cow was sedated using 2% xylazine hydrochloride (0.1 mg/kg) intravenously; next, the mouth was opened with a gag, and a veterinarian manually palpated the mass. The surface of the mass was inelastic on palpation, suggesting a thick wall and, thus, making it difficult to manually rupture it.

## 4. Treatment

The aim of the treatment was to incise the mass and extricate its contents; however, owing to its location and depth, accessing the mass from the body surface could have damaged the surrounding tissues. Hence, the mass was accessed from the mouth.

The patient was administered a sedative at a minimal dose of 2% xylazine hydrochloride (0.2 mg/kg), which was maintained throughout treatment. After sedative administration, the patient was placed in the right lateral recumbent position. A cuffed endotracheal tube was inserted directly into the trachea by manually displacing the pharyngeal abscess dorsally. Oxygenated isoflurane was administered to avoid dyspnea. The cow also received an intravenous injection of cefazolin sodium (5 mg/kg) to prevent perioperative infections and flunixin meglumine (2 mg/kg) for inducing analgesia.

The cow’s mouth was opened with a gag, and a veterinarian inserted a hand to reach the mass and palpate it; however, the mass was firm and could not be ruptured using manual pressure.

A rectal ultrasound probe (10.0 MHz linear probe, MyLab One VET, Esaote, Maastricht, The Netherlands) was inserted orally to identify the area with the minimum wall thickness of the mass and to assess the surrounding vascularity on Doppler ultrasonography.

An assistant inserted an endoscopic camera (8 mm) into the pharynx through the nasal cavity; then, the areas displayed on the monitor were examined and a thin-walled region of the mass with no blood vessels was identified on ultrasonography.

The space between the nasopharyngeal and mid-pharyngeal regions was narrow, allowing the insertion of a single human hand; furthermore, inserting a sharp tool, such as a scalpel, could have damaged the surrounding tissues. Therefore, the tip of a trocar was covered and manually inserted into the site displayed on the endoscopic monitor, and the mass was penetrated. No outflow of material was noted through the lumen of the trocar; thus, a finger was inserted to gradually widen the opening. Coagulated pus was carefully removed by compressing the mass with a finger. A portion of the material extracted was used for bacterial culture. Curettage was performed after pus removal, and saline solution was injected through the endoscope into the abscess for thorough washing.

Following curettage, the abscess cavity was washed with 2% povidone–iodine (Figure 4). The cow’s head was lowered intraoperatively to facilitate the drainage.

Recovery from anesthesia postoperatively was uneventful. Wheezing disappeared immediately after the surgery, and appetite was restored the following day. Follow-up endoscopies were performed through the nasal cavity, and the abscess cavity was washed with saline and 2% povidone–iodine for 3 days (Figure 5). Cefazolin sodium was also administered intravenously for 3 days postoperatively.

The pus was aseptically collected and subjected to aerobic and anaerobic cultures at 37 ℃ for 24 h in a 5% sheep blood agar medium. Anaerobic culture was performed in an anaerobic jar with Anero Pack (Mitsubishi Gas Chemical Co., Inc., Tokyo, Japan). The strains isolated from the aerobic and anaerobic cultures were identified as *Pseudomonas synxantha* and *Peptoniphilus indolious* using the MALDI biotyper (Bruker Daltonics, Billerica, MA, USA). A drug susceptibility test, which was performed as described by the Clinical and Laboratory Standards Institute, showed that *Pseudomonas synxantha* was sensitive to kanamycin and tetracycline, while *Peptoniphilus indolious* was sensitive to penicillin, cefazolin, tetracycline, and enrofloxacin.

## 5. Outcome and Follow-Up

The patient was discharged on postoperative day 4. Endoscopy 16 days postoperatively showed that the dorsal pharyngeal swelling had disappeared, and the airway patency had improved significantly (Figure 6). The owner reported that the cow had an uneventful recovery and was healthy even 1 year postoperatively. 

## 6. Discussion

Most causes of trauma or infection of the pharynx in cows are iatrogenic, predominantly relating to the improper manipulation of bolus-dosing tubes or other devices or owing to ingestion of magnetic or sharp objects by the cows. Symptoms may include marked swelling of the pharyngeal region, dyspnea, wheezing, drooling, and malodorous breathing [2,5]. In the present case, the cause of abscess formation was unknown, owing to the absence of abnormalities on the body surface and no history of drug administration or magnet ingestion.

Radiography, ultrasonography, and endoscopy have been proposed to diagnose upper respiratory tract obstructions [3]. Williams et al. used radiography and endoscopy to identify and manually crush abscesses [4]. They effectively confirmed the location of the abscess and obstruction of the trachea.

In this case, it was necessary to puncture the abscess using a tool because the thick abscess wall prevented rupturing it manually. Hence, it was necessary to accurately ascertain the thickness of the abscess wall and its vascularization. 

The linear probe of the ultrasonography system is rod-shaped and suitable for scanning abscesses in the pharynx within a small space. Computed tomography (CT) can detect lesions that cannot be diagnosed using radiography; however, it is limited to the head in calves, small ruminants, and adult cattle [3]. In addition, compared with these modalities, magnetic resonance imaging (MRI) may provide additional information for the diagnosis of abscesses in the posterior pharyngeal lymph nodes of cows, identifying the exact location and cause of the abscess [6]. However, currently, CT and MRI examinations for cattle pose several challenges, such as difficulty in transportation, economic constraints, and the need for general anesthesia.

Surgical methods in the case of diseases of the guttural pouch in horses, located on the dorsal side of the pharynx, include the Whitehouse approach, modified Whitehouse approach, approach through the Viborg’s triangle, and the hyovertebrotomy approach [7,8,9,10,11,12,13]. The modified Whitehouse approach facilitates good drainage [10], and a positive prognosis has been reported in a cow after a retropharyngeal abscess was cleaned using this approach [14]. However, it is associated with trauma to important nervous structures within the pouch walls, potentially resulting in dysphagia and cranial neuropathies [12]. The Whitehouse approach or modified Whitehouse approach requires avoidance of the spinal accessory, hypoglossal, glossopharyngeal, vagus, and facial nerves [10,15]. An incision in this area and blind manipulation can damage these important structures [16,17].

It is proven that if the abscess can be adequately cleaned and disinfected intraoperatively and postoperatively, systemic administration of antimicrobial agents is not necessary. Therefore, we opted for an oral approach. We considered it crucial to insert the incision instrument in a safe manner, as close to the abscess as possible in the confined space, while avoiding blood vessels, to incise and drain the abscess carefully.

Ultrasonography was useful in determining the appropriate abscess incision site for drainage, cleaning, and disinfection. Moreover, the combination of Doppler ultrasonography with endoscopy was ideal for determining the optimal puncture location because it provided information on wall thickness and vascularity.

Consequently, we were able to achieve adequate cleaning and disinfection of the abscess intraoperatively and for 3 days postoperatively, and no complications occurred, so no further antimicrobial agents were administered.

## 7. Conclusions

For deep abscesses, such as bovine pharyngeal abscesses, it is important to perform a preoperative transoral Doppler ultrasonography to assess the vascularity and thickness of the abscess wall for safe trocar insertion and drainage of abscess.

## Figures and Tables

**Figure 1 vetsci-09-00446-f001:**
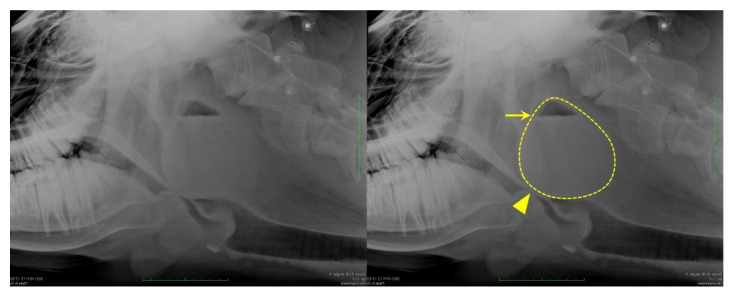
Lateral radiography of the pharyngeal area of the cow showing a retropharyngeal abscess obstructing the laryngeal airway (arrowhead). The dashed line indicates the mass outline, and the arrow points to the horizontal demarcating line, which indicates the presence of fluids.

**Figure 2 vetsci-09-00446-f002:**
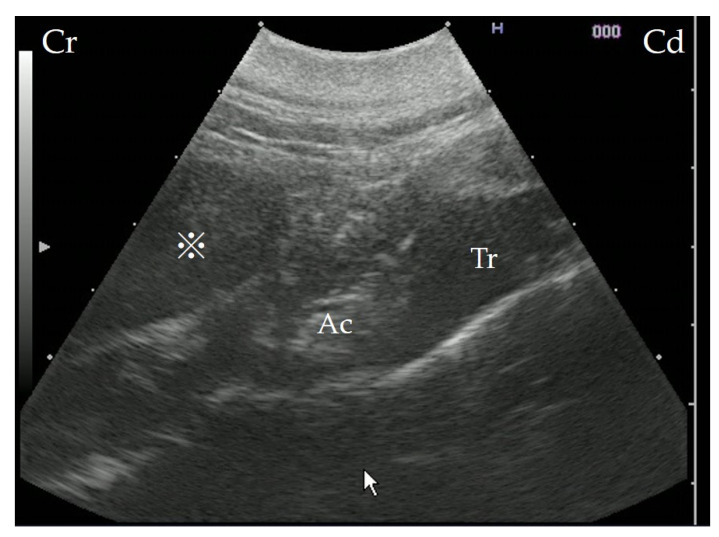
Ultrasonography showing the retropharyngeal mass. The scan was performed from the left side of the cow, with the probe in a transverse position; the trachea and epiglottic cartilage present as landmarks. A mass can be identified on the dorsal surface of the arytenoid cartilage. Cr: cranial; Cd: caudal; Ac: arytenoid cartilage; Tr: trachea area; ※: mass.

**Figure 3 vetsci-09-00446-f003:**
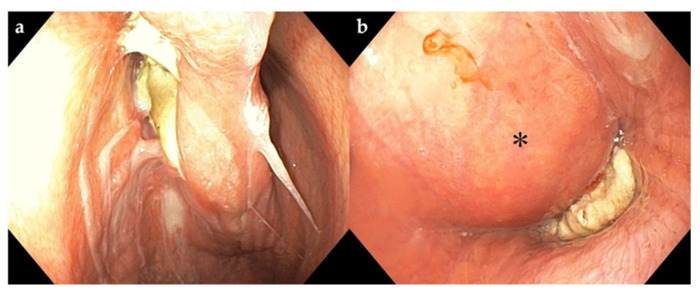
Endoscopic view of the dorsal aspect of the nasal passage and the upper respiratory tract with a retropharyngeal mass. (**a**) Pus-like material adherent to the dorsum of the nasal passage; (**b**) mass (*) in the pharyngeal region obstructing the airways.

**Figure 4 vetsci-09-00446-f004:**
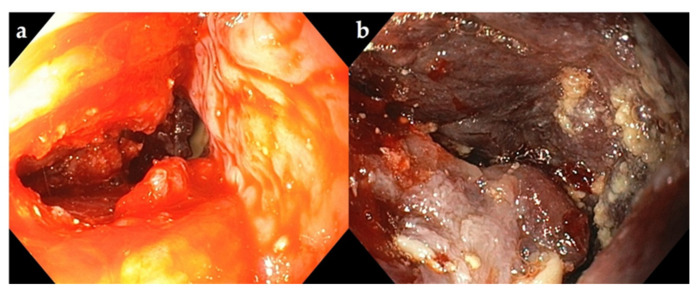
Endoscopic view of the pharyngeal abscess cavity (**a**) before and (**b**) after manual drainage and lavage.

**Figure 5 vetsci-09-00446-f005:**
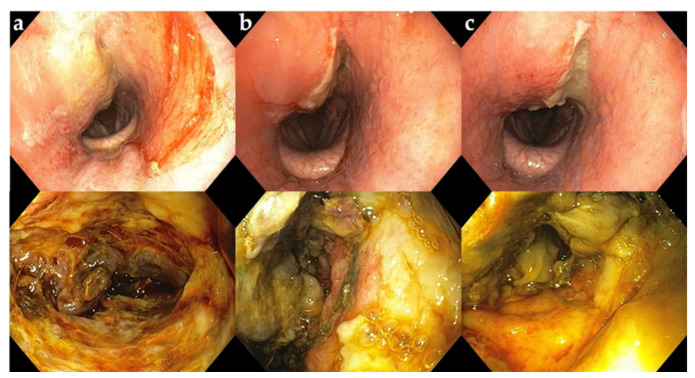
Endoscopic view of the upper respiratory area (upper row) and interior of the abscess (lower row) postoperatively after washing the cavity with povidone–iodine. (**a**) Day 1 after surgery; (**b**) day 2 after surgery; (**c**) day 3 after surgery.

**Figure 6 vetsci-09-00446-f006:**
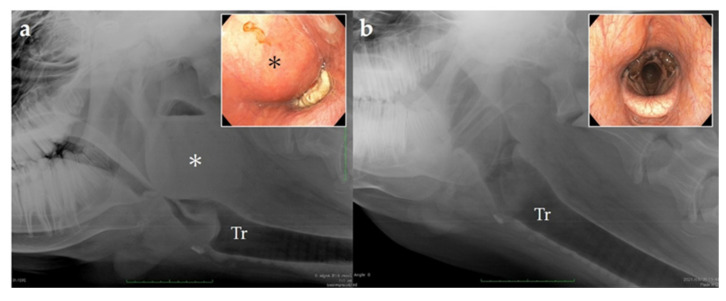
Lateral radiography of the pharyngeal area and endoscopic view via the nasal cavity. (**a**) Day of admission; (**b**) sixteen days after the surgery. Tr: trachea; *: abscess.

## Data Availability

All study data are presented in the article.

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
