# Peer review of "Surgical Treatment of a Retropharyngeal Abscess in a Japanese Black Cow"

_vetsci, 2022, doi:10.3390/vetsci9080446_

Round 1
Reviewer 1 Report
It´s a curious and interesting case report that proportionates clinical information for similar clinical cases giving good advice to readers.
Some corrections are necessary:
- line 164 "and the collected pus revealed Pseudomonas synxantha in aerobic cultures and Peptoniphilus indolious in anaerobic cultures" This sentence should be removed because is repeated.
The authors comment about antibiotic treatment, three days on cephazolin. After the microbiologic culture and its antibiogram 6 days after they explain the sensitivity of the antibiotics. But they do not describe if it was necessary to put more antibiotics or nor. Please comment and discuss the necessity to put more antibiotics or not.
- line 204 laryngeal sac in horses,
For better understanding, this sentence should be substituted by: gutural pouch in horses
Author Response
Point-by-Point Responses to Reviewer 1’s Comments
We appreciate the time and effort that the reviewer has dedicated for the meticulous assessment of our paper and for the pertinent comments. We have been able to incorporate changes to reflect most of the suggestions provided by the reviewers and have highlighted the changes within the manuscript in yellow. We have addressed each of the comments in a point-by-pint manner as follows.
Comment:
- line 164 "and the collected pus revealed Pseudomonas synxantha in aerobic cultures and Peptoniphilus indolious in anaerobic cultures" This sentence should be removed because is repeated.
[Response]
We are grateful for the meticulous review of our manuscript. As suggested by the reviewer, this sentence has been deleted from the revised manuscript.
- The authors comment about antibiotic treatment, three days on cephazolin. After the microbiologic culture and its antibiogram 6 days after they explain the sensitivity of the antibiotics. But they do not describe if it was necessary to put more antibiotics or nor. Please comment and discuss the necessity to put more antibiotics or not.
[Response]
We thank the reviewer for this constructive comment. We believe that systemic administration of antimicrobials is unnecessary in the event that the abscess can be adequately cleaned and disinfected, which was the case in the present study; therefore, we prophylactically administered systemic antibiotics for 3 days. After the abscess was sufficiently cleaned and disinfected for 3 days after surgery, administration of further antimicrobials was discontinued.
 This information has been incorporated in the Discussion section in Lines 225–236 of the revised manuscript.
- line 204 "laryngeal sac in horses, "
For better understanding, this sentence should be substituted by: gutural pouch in horses
[Response]
We thank the reviewer for pointing this out. Accordingly, “laryngeal sac” has been corrected to “guttural pouch” in Line 214 of the revised manuscript.

Reviewer 2 Report
General comment: The authors presented an interesting report of a retropharyngeal abscess in a Japanese black cow.The manuscript should be revised by an English native speaker.
Title: The title short, clear, and concise.
Abstract: It is adequate.
The keywords should be different from those used in the title.
Introduction: It is adequate. The authors provided an adequate overview of the thematic.
Methods: The methods should be improved.
How were the blood samples collected?
How was the bacteriological analysis performed?
Was the animal sedated or anesthetized for radiographic and ultrasonographic analysis.
Results: They are presented properly and supported by the Figures.
Discussion: It is adequate.
Conclusion: The conclusion is adequate.
References: Adequate.
Recommendation: The manuscript should be accepted for publication after a Moderate revision.
Author Response
Point-by-Point Responses to Reviewer 2’s Comments:
We appreciate the time and effort that the reviewer has dedicated for the meticulous assessment of our paper and for the pertinent comments. We have been able to incorporate changes to reflect most of the suggestions provided by the reviewers and have highlighted the changes within the manuscript in yellow. We have addressed each of the comments in a point-by-pint manner as follows.
Comment:
How were the blood samples collected?
[Response]
Blood was drawn from the jugular vein, and we have incorporated this information in the revised manuscript in Lines 70–75.
How was the bacteriological analysis performed?
[Response]
Methods of bacteriological analysis were added to the text in Lines 170–175 of the revised manuscript.
Was the animal sedated or anesthetized for radiographic and ultrasonographic analysis.
[Response]
Radiography, ultrasonography and endoscopy were performed without anesthesia or sedation. This information has been added, as pointed out by the reviewer, to Lines 82, 94, and 120 of the revised manuscript.

Author Response
Point-by-Point Responses to Reviewer 3’s Comments:
We thank the reviewer for the meticulous assessment of our paper and for the pertinent comments.

Round 2
Reviewer 2 Report
The manuscript should be accepted for publication in the present form.